# Unveiling User Satisfaction and Creator Productivity Trade-Offs in Recommendation Platforms

**Fan Yao**[*]
University of Virginia
fy4bc@virginia.edu

**Yiming Liao**
Meta
yimingliao@meta.com

**Jingzhou Liu**
Meta
jingzhol@meta.com

**Shaoliang Nie**
Meta
snie@meta.com

**Qifan Wang**
Meta
wqfcr@meta.com

**Haifeng Xu**
University of Chicago
haifengxu@uchicago.edu

**Hongning Wang**
University of Virginia
hw5x@virginia.edu

## Abstract

On User-Generated Content (UGC) platforms, recommendation algorithms significantly impact creators' motivation to produce content as they compete for algorithmically allocated user traffic. This phenomenon subtly shapes the volume and diversity of the content pool, which is crucial for the platform's sustainability. In this work, we demonstrate, both theoretically and empirically, that a purely relevance-driven policy with low exploration strength boosts short-term user satisfaction but undermines the long-term richness of the content pool. In contrast, a more aggressive exploration policy may slightly compromise user satisfaction but promote higher content creation volume. Our findings reveal a fundamental trade-off between immediate user satisfaction and overall content production on UGC platforms. Building on this finding, we propose an efficient optimization method to identify the optimal exploration strength, balancing user and creator engagement. Our model can serve as a pre-deployment audit tool for recommendation algorithms on UGC platforms, helping to align their immediate objectives with sustainable, long-term goals.

## 1 Introduction

User-generated content (UGC) platforms have become an indispensable component of our daily lives [5, 45]. Those platforms, including various social media (e.g., Facebook, Instagram), streaming services (e.g., YouTube, TikTok) and many more, count on algorithmic recommendation algorithms [36, 5] to help content consumers (i.e., users) navigate the vast ocean of content generated by creators. Unlike other content recommendation platforms such as Netflix and Spotify, user experience on UGC platforms critically relies on the active participation of creators [58], as the goal of enhancing user engagement is inherently linked to the abundancy and diversity of the content pool.

Recent studies have begun to explore how a platform's algorithmic decisions, such as their employed recommendation algorithms and revenue sharing agreements, might influence the behavior of content creators and subsequently affect user welfare [52, 54, 55, 34, 30, 33, 32, 43]. A common technique used in these works is to model the competition among creators who strive to establish their brand and comparative advantages by selecting topics that maximize the traffic or rewards from the platform. While these competition models offer valuable insights into how platforms' interventions could lead to suboptimal outcomes in terms of content diversity and issues related to popularity bias, they ignore the dimension of content creation volume — an equally critical aspect of such competition

---

[*]Research conducted during an internship at Meta. Corresponding author.

38th Conference on Neural Information Processing Systems (NeurIPS 2024).

dynamics. In fact, there are evidence suggesting that traffic received directly influences creator productivity [31, 56]. A recent study on a leading video recommendation platform found that creators whose content received boosted exposure significantly increased their video production without compromising quality [31]. And a recent field experiment on a large-scale video sharing social network showed that while popularity-based recommendation strategies boost content consumption, they can reduce content production [56]. The similar effect has also been documented on Instagram. For instance, when the recommender system allocates more traffic to targeted groups of creators, their production frequency increases, as indicated by metrics such as the number of daily active creators and the daily average creation volume per group. However, disproportionately boosting traffic to certain creator groups can negatively impact others; for instance, directing more traffic to popular, or "head", creators often diminishes engagement from less prominent, or "tail", creators. And overall, a statistically significant positive correlation exists between content viewership and the corresponding creator's productivity[2].

Motivated by real-world evidence and the research gap in modeling production frequency within content creation competition, we introduce a new game-theoretical model, named Cournot Content Creation Competition ($C^4$), aiming to study the impact of a platform's recommendation strategy on creators' production willingness. Our $C^4$ framework builds upon the Content Creator Competition ($C^3$) framework introduced in [52, 54, 55], assuming that creators compete for platform-allocated user traffic [27, 29]. However, unlike $C^3$ and similar previous models, our approach models the competition where creators are aware of their expertise and consistently produce within their niche, but strategically adjust their production frequency to balance gain and cost, which is often observed on mature platforms such as YouTube and Instagram. Our proposed competition model resonates with the well-established Cournot competition [14] in economics, where firms compete for revenue by strategically setting their production quantities. Hence, it inherits its name.

Our $C^4$ framework offers a powerful tool for analyzing the competition dynamics among creators, as it always yields a unique Pure Nash equilibrium (PNE) that enables a precise prediction of the total content creation volume under any specific recommendation strategy. Furthermore, our in-depth analysis of $C^4$'s equilibrium reveals a critical and interesting insight: while increased recommendation accuracy boosts immediate user satisfaction, it simultaneously reduces creators' motivation to produce content, potentially compromising long-term user engagement. This finding, supported by both theoretical analyses and simulations, suggests the necessity to balance user and creator engagement through a careful control over the recommendation algorithm's exploration strength at a per-user basis. We formulated this mechanism design challenge as a bi-level optimization problem and tackled it using a projected gradient descent approach with an efficient gradient approximation scheme, providing an effective method to achieve the optimal trade-off between user satisfaction and creator productivity.

In summary, our contributions are threefold: (a) **modeling-wise**, we introduce a new game-theoretical framework, $C^4$, to investigate how recommendation algorithms affect content creation frequency among creators; (b) **conceptually**, we reveal a new insight that managing the exploration strength of the recommendation algorithm can balance between short-term user satisfaction and long-term creator engagement at equilibrium; and (c) **technique-wise**, we reformulate the mechanism design problem of identifying the optimal engagement trade-off at the equilibrium into a solvable offline optimization problem, tackled using approximated gradient descent. Our $C^4$ framework and its derived solution serve as a pre-deployment audit tool for platforms, assessing the effects of algorithmic choices on creator and user engagement.

## 2 Related Work

The study of online content creation economy has captured the attention of machine learning community recently, leading to a diverse collection of models addressing the dynamics of content creator competition [25, 24, 3, 54, 52, 57, 32, 34, 30, 16]. In these models, creators strategically select the type [49], topic [30, 34, 52], or quality [25, 32] of their content, competing for resources such as traffic [30, 2, 24], user engagement [52], or platform-provided incentives [57, 54]. Some models aim to explore the properties of creator-side equilibrium, investigating how creators specialize at

---

[2]Further details regarding the Instagram findings are protected under a non-disclosure agreement (NDA) and may be disclosed in future versions of this work.

equilibrium [34], the impact of creators' strategic behaviors on social welfare [52], and the design of optimization methods for long-term welfare considering these behaviors [2, 3, 54, 57, 32, 33, 39]. Unlike these works, which generally assume equally paced creation frequency among creators, our proposed $C^4$ games consider scenarios where creators strategically control their creation quantities, allowing us to analyze how recommendation algorithms influence overall content creation volume at the equilibrium.

Our model of the platform's recommendation algorithm draws inspiration from the proportional allocation concept in game theory, applicable to resource distribution [9, 41] and contest design [48]. We assume that each user contributes a unit of traffic, which is allocated to creators based on both of their merit and effort (content creation frequency). This modeling is closely related to the Tullock contest [48], also known as the lottery contest, where the probability of winning a fixed prize is proportionate to the effort expended relative to the total effort by all contestants. While the Nash equilibrium of the one-dimensional Tullock contest with homogeneous costs is well-understood [20, 21], our work extends this framework to include heterogeneous convex costs. A recent study [53] also explored competition between human and Generative AI creators within a similar setup, examining its impact on total content creation volume. However, they did not address the influence of different recommendation algorithms, which we investigate in our work.

In a broader sense, our work contributes to a line of research evaluating the impact of recommender systems on individuals, specifically exploring how deployed algorithms shape user and content creator behavior [16, 12, 11, 38, 17, 35, 19] and how we can design new algorithms to address these effects [10, 50, 6, 51, 4, 43, 1]. Our study introduces a key insight: recommender algorithms optimized solely for user satisfaction can unintentionally reduce content creators' willingness to engage, thereby impacting long-term user engagement. We address this challenge by proposing a solution that balances creator engagement and user satisfaction through imposing exploration strengths tailored to individual users.

## 3   The Cournot Content Creation Competition

In this section, we introduce the formulation of Cournot Content Creation Competition ($C^4$), which models the competition among creators for traffic on UGC platforms. This model considers the potential impact of the platform's traffic reallocation mechanisms, such as their deployed recommendation algorithms, where creators strategically choose their production frequencies to optimize their allocated traffic. Each $C^4$ game instance $\mathcal{G}$ is characterized by a tuple $(n, m, M, \{c_i\}_{i=1}^n, \{\beta_i\}_{i=1}^m)$. We detail each component of this tuple as follows:

1. **Basic setups.** There is a set of content creators denoted by $[n] = \{1, \cdots, n\}$, and a set of users denoted by $\{u_j\}_{j=1}^m$. We assume each user $j$ has a stable preference over creators and such relationship is captured by an $n$-by-$m$ matrix $M$ with its $(i, j)$-th entry $w_{ij} \in [0, 1]$ denoting the strength of user $j$'s preference over creator $i$'s content. Each creator determines a production frequency $x_i \in \mathbb{R}_{\geq 0}$ in a unit amount of time (e.g., one week/one month). For the purpose of our analysis, $x_i$ can be interpreted interchangeably as either the production frequency or volume, provided there is no ambiguity[3]. Follow the terminology of game theory literature, $x_i$ is referred to as the action or pure strategy of creator $i$. Each creator $i$ is associated with a cost function $c_i$, which characterizes the cost for creating content at frequency $x_i$. We have two assumptions about $c_i$: 1. $c_i$ is increasing in $x_i$ and $c_i(x_i) \to +\infty$ as $x_i \to +\infty$, which reflects that content creation is always not free. 2. $c_i$ is convex in $x_i$, indicating a non-decreasing marginal cost of improving production frequency.

2. **Platform intervention.** We assume that each user $u_j$ contributes a unit amount of traffic, and the platform redistributes the total user traffic based on a relevance-based recommendation algorithm that adheres to certain probabilistic principle. Specifically, the recommendation algorithm matches $u_j$ to each piece of content produced by creator $i$ with a probability proportional to $\exp(\beta_j \sigma_{ij})$, where $\sigma_{ij} = w_{ij} + \epsilon_{ij}$ represents the algorithm's estimated relevance score. This score combines the true preference score $w_{ij}$ with independent Gaussian noise $\epsilon_{ij} \sim \mathcal{N}(0, \sigma^2)$. The parameter $\beta_j \geq 0$ governs the exploration of the matching for user $u_j$: a higher $\beta_j$ results in a more precise

---

[3]For the elegance of our theoretical analysis, we treat $x_i$ as a continuous variable, although the key messages and insights of this paper are preserved if $x_i$ is discrete.

matching, while a lower $\beta_j$ introduces more exploration into the matching results [4]. In this work, we analyze intervention mechanisms that operate under this Personalized Probabilistic Matching (PPM) principle, parameterized by $\boldsymbol{\beta} = \{\beta_i\}_{i=1}^m$, and refer to it as PPM($\boldsymbol{\beta}$) for ease of notations.

3. **Creator utility.** Creators are reward-seeking individuals who try to maximize the expected traffic for their created content while carefully balancing the costs. Under PPM($\boldsymbol{\beta}$) and when each creator $i$ produces $x_i$ copies of content, the platform will allocate to $i$ the amount of traffic from $u_j$ proportional to $\mathbb{E}_{\epsilon_{ij} \sim \mathcal{N}(0,\sigma^2)} \left[ x_i e^{\beta_j(w_{ij}+\epsilon_{ij})} \right] = x_i e^{\beta_j w_{ij}} \cdot e^{\frac{\sigma^2 \beta_j^2}{2}}$. Here we should note that $x_i$ does not suggest the creator would create $x_i$ pieces of identical content, but amount of content following his/her expertise. Therefore, we formulate creator $i$'s utility function as the following:

$$u_i(x_i, \boldsymbol{x}_{-i}; \boldsymbol{\beta}) = \sum_{j=1}^m \left( \frac{\mathbb{E}[x_i e^{\beta_j \sigma_{ij}}]}{\sum_{k=1}^n \mathbb{E}[x_k e^{\beta_j \sigma_{kj}}]} \right) - c_i(x_i)$$

$$= \sum_{j=1}^m \left( \frac{x_i e^{\beta_j w_{ij}}}{\sum_{k=1}^n x_k e^{\beta_j w_{kj}}} \right) - c_i(x_i), \tag{1}$$

where $\boldsymbol{x}_{-i} \in \mathbb{R}_{\geq 0}^{n-1}$ denotes the strategy profile of all creators except $i$.

The $C^4$ game models the scenario where creators are aware of their expertise (i.e., what topic to create), and compete purely on *creation quantity*. This concept is akin to the extensively studied Cournot competition [14] model in economics, where firms independently determine the output of homogeneous products at different costs. However, our model diverges from the classic Cournot competition in several key aspects. First of all, they have different revenue functions: in Cournot competition, the revenue for each firm is calculated as the product of price and production quantity, and the price only depends on all firms' joint strategy. In contrast, in the $C^4$ game, the gain of a creator not only relies on other creators' decision but also hinges on the platform's traffic allocation algorithm. In addition, while Cournot competition typically incorporates only linear cost functions, the $C^4$ game accommodates general convex cost functions, offering a more nuanced reflection of the real cost faced by creators. These distinctions highlight the unique aspects of our model, while maintaining a conceptual link to traditional economic theories of competition.

**Research questions:** Under the $C^4$ framework, an important and natural research question is how we can predict creators' strategic choices in the competition. This is a fundamental question in game theory and we employ the concept of Pure Nash Equilibrium (PNE) [40] to characterize the outcome of $C^4$ games. The definition of PNE is given by the following:

**Definition 1.** *A joint strategy profile of all creators $\boldsymbol{x}^* = (x_1^*, \cdots, x_n^*)$ forms a pure Nash equilibrium (PNE), if for every creator $i$, $x_i^*$ is a best response strategy that maximizes $u_i$ given other creators' strategy $\boldsymbol{x}_{-i}$; formally,*

$$u_i(x_i^*, \boldsymbol{x}_{-i}^*) \geq u_i(x_i, \boldsymbol{x}_{-i}^*) \ \text{for every } x_i \in \mathbb{R}_{\geq 0}, \forall i \in [n]. \tag{2}$$

In other word, a PNE represents a stable state when everyone is satisfied with their strategies and does not want to deviate. As we will demonstrate in the subsequent section, a PNE always exists in $C^4$ under any PPM($\boldsymbol{\beta}$) and can be computed efficiently. This finding forms the basis for our further theoretical analysis and empirical simulations.

In addition to the predictability and stability of creators' production strategies, it is equally crucial for platform designers to develop metrics that encourage the prosperity of the content ecosystem. They must also devise algorithmic solutions to optimize these metrics, balancing the trade-off between engagement of users and creators, using the available "knob" PPM($\boldsymbol{\beta}$). We will explore these issues in the upcoming technical discussions.

## 4 The PNEs of $C^4$ Games and Their Properties

As widely known, the PNE does not always exist [18, 23, 26]. However, our first main result establishes that, under mild assumptions, $C^4$ always admits a unique PNE.

---

[4]The randomness in matching results may stem from either the imperfect estimation of the preference score $w_{ij}$ or intentionally injected exploration strength based on the intervention mechanism. In this paper, we focus on the latter source of randomness and analyze how such exploration strength might affect the outcomes.

**Theorem 1.** *For any $C^4$ instance $\mathcal{G}(n, m, M, \{c_i\}_{i=1}^n, \boldsymbol{\beta})$. If each $c_i$ is convex in $x_i$, $\mathcal{G}$ admits a unique PNE.*

Note that the primary challenge in proving Theorem 1 is to show $\mathcal{G}$ is a strictly monotone game, whereas the existence and uniqueness of PNE in such games is a classic result from [44]. Theorem 1 is interesting from multiple perspectives. First, it strictly generalizes previous equilibrium existence results in classic Tullock contest [42, 13], which corresponds to the special case when $m = 1$ and $w_{11} = \cdots = w_{n1}$. Second, the fact that $C^4$ games are monotone is significant because it is well-known that the PNE of strictly monotone games can be found efficiently. For example, many natural multi-agent online learning dynamics such as mirror descent [7], accelerated optimistic gradient [8], and payoff-based learning [47] guarantee the last-iterate convergence to the unique PNE in strictly monotone games, even when players have mere zeroth order feedback about their utility functions. These results suggest that the PNE of $\mathcal{G}$ is achievable if all creators use a reasonable update rule in their strategies. This observation not only makes this equilibrium a plausible prediction of real-world competition but also paves the way to our simulation-based studies in our experiments, where we use multi-agent mirror descent with perfect gradient to numerically solve the PNE of $C^4$.

In addition to the existence and uniqueness properties, the following corollary characterizes the first-order characterization of $\mathcal{G}$'s PNE.

**Corollary 1.** *The unique PNE $\boldsymbol{x}^*(\boldsymbol{\beta})$ of any $\mathcal{G}(n, m, M, \{c_i\}_{i=1}^n, \boldsymbol{\beta})$ satisfies the following first-order condition:*

$$\left. \frac{\partial u_i}{\partial x_i} \right|_{\boldsymbol{x} = \boldsymbol{x}^*(\beta)} = 0, \quad 1 \le i \le n. \tag{3}$$

Since we have already shown the existence and uniqueness of $\mathcal{G}$'s PNE, Corollary 1 follows immediately according to Definition 1. Corollary 1 is useful for establishing further properties of $C^4$ games in Section 5.

## 5 The Trade-Off Between User and Creator Engagement

We have established that a unique PNE exists in any $C^4$ game under any PPM($\boldsymbol{\beta}$) and can be naturally achieved by competing content creators. This raises a crucial question for platform designers: how should the quality of the PNE be evaluated? For any mature UGC platform, it is essential to balance user satisfaction, which is key to short-term prosperity, with creator engagement, which is crucial for long-term sustainability. Within our $C^4$ framework, this requires the platform designer to generate matching results that not only guarantee high user satisfaction (by improving the average matching quality at PNE) but also stimulate substantial content creation volume (by encouraging creators to increase their production frequency $x^*$ at PNE). We define these two objectives as follows:

**Definition 2.** *For any $C^4$ instance $\mathcal{G}(n, m, M, \{c_i\}_{i=1}^n, \boldsymbol{\beta})$, let $\boldsymbol{x}^*(\boldsymbol{\beta}) = (x_1^*, \cdots, x_n^*)$ be the unique PNE under PPM($\boldsymbol{\beta}$). Then the (short-term) total user satisfaction is defined as*

$$U(\boldsymbol{x}^*(\boldsymbol{\beta}); \boldsymbol{\beta}) = \sum_{j=1}^m \sum_{i=1}^n \left( \frac{w_{ij} x_i^* e^{\beta_j w_{ij}}}{\sum_{k=1}^n x_k^* e^{\beta_j w_{kj}}} \right), \tag{4}$$

*and the (long-term) total content creation volume is defined as*

$$V(\boldsymbol{x}^*(\boldsymbol{\beta})) = \sum_{i=1}^n x_i^*. \tag{5}$$

*The social welfare of the whole system is measured by a linear combination of $U$ and $V$, defined as*

$$W_\lambda(\boldsymbol{x}^*(\boldsymbol{\beta}); \boldsymbol{\beta}) = U(\boldsymbol{x}^*(\boldsymbol{\beta}); \boldsymbol{\beta}) + \lambda V(\boldsymbol{x}^*(\boldsymbol{\beta}); \boldsymbol{\beta}). \tag{6}$$

If we denote $\pi_j^*(\boldsymbol{x}^*(\boldsymbol{\beta}), \boldsymbol{\beta}) = \sum_{i=1}^n \left( \frac{w_{ij} x_i^* e^{\beta_j w_{ij}}}{\sum_{k=1}^n x_k^* e^{\beta_j w_{kj}}} \right)$ as the indicator of an individual user's satisfaction or utility, which is the expected matching scores of user $j$ at the PNE under PPM($\boldsymbol{\beta}$). And the total user satisfaction measure $U = \sum_{j=1}^m \pi_j^*$ is the accumulated user utility. We argue that $U$ primarily serves as a metric for short-term welfare evaluation, since it focuses solely on user satisfaction at a specific instance of matching outcomes but does not capture the dynamics of user

engagement over time. This overlooks the crucial fact that sustained user engagement on a platform requires a continuous supply of relevant content, as users can hardly be satisfied by their previously consumed material. This limitation is also evident in $U$'s mathematical formulation: its value remains unchanged with a rescaling of $\{x_i^*\}$, indicating that it fails to reflect changes in content volume or frequency that might affect long-term user engagement. On the other hand, the long-term prosperity of a UGC platform is fundamentally linked to the engagement of content creators. Therefore, we introduce the total content creation volume $V$ as an indicator of the long-term welfare of the platform.

For the platform designers, it is essential to develop metrics that balance both short-term and long-term considerations. Thus, we propose a hybrid social welfare metric, $W$, which combines $U$ and $V$ to reflect both user satisfaction and content supply sustainability. However, understanding the mechanisms to optimize $U$ and $V$ independently is critical. In the following sections, we will explore the optimal matching mechanisms tailored to the exclusive objectives of $U$ (short-term) and $V$ (long-term), and then present an efficient algorithm designed to optimize $W$.

Interestingly, our findings suggest that both $U$ and $V$ exhibit monotonicity with respect to the parameter $\beta$, even in scenarios involving a homogeneous user population. This uniform behavior of $U$ and $V$ offers valuable insights into how the adjustment of exploration strength could potentially impact platform performance. Our forthcoming theorem formally characterizes these observations.

**Theorem 2.** *Consider any $C^4$ game with $m = 1$. If the elements of $M = [w_1, \cdots, w_n]^\top$ are not identical, it holds that:*

1. *$U(\beta)$ defined in Eq. (4) is strictly increasing in $\beta$.*

2. *$V(\beta)$ defined in Eq. (5) is strictly decreasing in $\beta \in [\beta_0, +\infty)$ for some $\beta_0 > 0$.*

Theorem 2 conveys two significant insights. The first one, though perhaps unsurprising, reveals that improving matching accuracy corresponds to an increase in expected user satisfaction. Despite its intuitiveness, this is a strong observation because it holds without relying on any specific structural assumptions about creator cost functions. This means that regardless of the potential complexity in equilibrium structures due to creator costs, and even when the order of $x_i^*$ does not align with a creator's capability $w_i$, the metric $U$ is still monotonically increasing with respect to $\beta$. The second insight may surprise some readers: it suggests that while keep increasing the matching accuracy motivates some creators to produce more content, it demotivates others, resulting in a net decrease in the overall volume of content creation. This finding illustrates an intrinsic trade-off between short-term matching accuracy and long-term content supply: strategies that enhance short-term user satisfaction can inadvertently reduce content creation frequency across creators. To the best of our knowledge, this result is novel and has not been discussed in similar studies.

Here is an intuitive explanation for why a large $\beta$ diminishes creators' willingness to produce content. As the traffic allocation becomes more deterministic, the marginal gain from increasing production frequency diminishes because the amount of traffic accrued is largely determined by the relevance score, rather than volume. In the extreme case where $\beta \to +\infty$, only the most relevant creator captures all the user traffic, regardless of her production volume. Consequently, due to the presence of production costs, this creator, and others, will only sustain the minimum viable productivity. Conversely, in the other extreme scenario where $\beta = 0$, i.e., user traffic is distributed uniformly among creators irrespective of relevance, the gain for each creator depends solely on their production frequency, prompting a productivity arms race. Clearly, both extremes are suboptimal, but they effectively illustrate the rationale behind our theoretical findings.

Although in Theorem 2 we consider the game instance $\mathcal{G}$ with $m = 1$ as a representative snapshot of how creators compete for a single unit of user traffic (i.e., homogeneous user population), extending the time frame to encompass a sequence of heterogeneous users suggests that the observed trade-off between $U$ and $V$ remains consistent. In our experiments, we will demonstrate this trade-off in broader settings through simulations, e.g. when $m > 1$ and with various complex user distributions.

The proof of Theorem 2, while delivering a clear message, is far from trivial. Since the dependencies of $U$ and $V$ on $\beta$ are indirectly linked through $x^*$, which lacks a closed form, the derivation of their derivatives with respect to $\beta$ necessitates the use of the implicit function theorem [37] to articulate the derivative of $x^*$ with respect to $\beta$. This involves a complex matrix inverse, which we simplify using the Sherman–Morrison formula [46] due to its structure being a diagonal matrix with a rank-one update. This proof technique not only supports our theorem but also inspires a novel first-order

optimization approach to address the hybrid social welfare optimization discussed in Section 7. Detailed proofs are provided in Appendix A.2.

# 6 Finding the Optimal Trade-off through Optimization

Our theory thus far indicates that optimizing both user satisfaction $U$ and creator engagement $V$ is non-trivial, even when the user population is homogeneous, as achieving the optimal of $U$ and $V$ simultaneously is impossible. Consequently, an essential and intriguing question arises within any specific competitive environment $C^4$: how can we identify the optimal trade-off between these two factors by optimizing any given welfare metric $W_\lambda$?

Generally, the welfare metric $W$ is influenced by three factors: the platform's algorithmic recommendation policy $\text{PPM}(\boldsymbol{\beta})$, the resulting content creation profile $\boldsymbol{x}^*$ at the PNE induced by $\boldsymbol{\beta}$, and the relevance matrix $M$. Thus, we can formulate the resulting optimization problem (OP) as follows:

$$\text{Find} \quad \arg\max_{\boldsymbol{\beta} \in \mathbb{R}^m_{\geq 0}} W_\lambda(\boldsymbol{x}^*(\boldsymbol{\beta}), \boldsymbol{\beta}) \tag{7}$$

$$s.t. \quad \boldsymbol{x}^*(\boldsymbol{\beta}) \text{ is the PNE of } \mathcal{G}.$$

In general, OP (7) presents a formidable challenge, as solving for a PNE of a game is known to be difficult [15]. Fortunately, the nice structure of $C^4$ allows us to utilize the implicit characterization of the PNE detailed in Corollary 1 to tackle OP (7) effectively. In the subsequent section, we demonstrate that the gradient of $W_\lambda$ w.r.t. $\boldsymbol{\beta}$ can be explicitly computed.

## 6.1 The Derivation of Exact Gradient

According to the chain rule, the first-order gradient of $W_\lambda$ w.r.t. $\boldsymbol{\beta}$ can be expressed as

$$\frac{dW_\lambda}{d\boldsymbol{\beta}} = \frac{dU(\boldsymbol{x}^*(\boldsymbol{\beta}), \boldsymbol{\beta})}{d\boldsymbol{\beta}} + \lambda\frac{dV(\boldsymbol{x}^*(\boldsymbol{\beta}))}{d\boldsymbol{\beta}} = \left(\frac{\partial U}{\partial \boldsymbol{x}^*} + \lambda\frac{\partial V}{\partial \boldsymbol{x}^*}\right) \cdot \frac{d\boldsymbol{x}^*}{d\boldsymbol{\beta}} + \frac{\partial U}{\partial \boldsymbol{\beta}}. \tag{8}$$

The evaluation of the gradient of $W_\lambda$ relies on the calculation of three vectors, $\frac{\partial U}{\partial \boldsymbol{x}^*}, \frac{\partial V}{\partial \boldsymbol{x}^*} \in \mathbb{R}^{1\times n}$, and $\frac{\partial U}{\partial \boldsymbol{\beta}} \in \mathbb{R}^{1\times m}$, as well as a Jacobian matrix $\frac{d\boldsymbol{x}^*}{d\boldsymbol{\beta}} \in \mathbb{R}^{n\times m}$. The computations of $\frac{\partial U}{\partial \boldsymbol{x}^*}, \frac{\partial V}{\partial \boldsymbol{x}^*}$ and $\frac{\partial U}{\partial \boldsymbol{\beta}}$ are straightforward and computationally light, which position the main challenge as the computation of $\frac{d\boldsymbol{x}^*}{d\boldsymbol{\beta}}$. Fortunately, the first-order characterization of $\boldsymbol{x}^*$ by Corollary 1 enables us to express the gradient of $\boldsymbol{x}^*$ w.r.t. $\boldsymbol{\beta}$ using implicit function derivation [37] as $\frac{d\boldsymbol{x}^*}{d\boldsymbol{\beta}} = -\left(\frac{\partial F}{\partial \boldsymbol{x}^*}\right)^{-1} \cdot \frac{\partial F}{\partial \boldsymbol{\beta}}$, where $F(\boldsymbol{x}, \boldsymbol{\beta}) = \left(\frac{\partial u_i}{\partial x_i}\right)^n_{i=1}$ is an $n$-valued function, and both $\frac{\partial F}{\partial \boldsymbol{x}^*}$ and $\frac{\partial F}{\partial \boldsymbol{\beta}}$ are matrices of dimensions $n \times n$ and $n \times m$, respectively. The following proposition provides the exact formula for the gradient. The calculation is straightforward and we omit the detailed derivation.

**Proposition 1.** *Let $\boldsymbol{x}^* = (x_1^*, \cdots, x_n^*)$ be the PNE of $\mathcal{G}(n, m, M, \{c_i\}_{i=1}^n, \boldsymbol{\beta})$. Then, the Jacobian matrix of $\boldsymbol{x}^*$ as a function of $\boldsymbol{\beta}$ is*

$$\frac{d\boldsymbol{x}^*}{d\boldsymbol{\beta}} = \left(D + YZ^\top\right)^{-1} B \in \mathbb{R}^{n\times m}, \tag{9}$$

*where $D$ is an $n \times n$ diagonal matrix given by*

$$D = diag\left(c_1'' + \sum_{j=1}^m \frac{P_{1j}^2}{x_1^{*2}}, \cdots, c_n'' + \sum_{j=1}^m \frac{P_{nj}^2}{x_n^{*2}}\right), \tag{10}$$

*and $B, Y, Z$ are $\mathbb{R}^{n\times m}$ matrices calculated as follows ($1 \leq i \leq n, 1 \leq j \leq m$):*

$$Y = \left[\frac{P_{ij}(1 - 2P_{ij})}{x_i^*}\right]_{ij}, \quad Z = \left[\frac{P_{ij}}{x_i^*}\right]_{ij}, \quad B = \left[\frac{P_{ij}(1 - 2P_{ij})}{x_i^*} \cdot \left(w_{ij} - \sum_{k=1}^n w_{kj}P_{kj}\right)\right]_{ij}, \tag{11}$$

*where $c_i''$ is the second-order derivative of creator $i$'s cost function, and $P_{ij} = \frac{x_i^* \exp(\beta_j w_{ij})}{\sum_{k=1}^n x_k^* \exp(\beta_j w_{kj})}$ is the probability that creator $i$ is matched with user $j$ at the PNE under $PPM(\boldsymbol{\beta})$.*

## 6.2 Optimization with Approximated Gradients

Proposition 1 together with Eq. (8) offers us a possibility to directly apply gradient-based approaches for solving OP (7). However, the gradient computation requires the inversion of an $n \times n$ matrix, whose time complexity is $O(n^3)$ and thus too cumbersome. To reduce the computational burden, we propose to approximately compute the gradient using the Sherman–Morrison-Woodbury formula [46] to approximate the matrix inverse, inspired by the specific structure of the RHS of Eq. (9). According to Sherman–Morrison-Woodbury formula, it holds that

$$(D + Y Z^\top)^{-1} = D^{-1} - D^{-1} Y \left( I + Z^\top D^{-1} Y \right)^{-1} Z^\top D^{-1}, \tag{12}$$

and the computation of the RHS of Eq. (12) now requires a time complexity of $O(n^2 m + n m^2 + m^3)$. However, the size of the user population $m$ in practical scenarios is often even larger than $n$. To efficiently compute the RHS of Eq. (12), we propose a method to "sketch" the matrices $Y$ and $Z$ by sampling a subset of users. Initially, each column of $Y$ and $Z$ corresponds to a user index $j$. We begin by sampling a sub-population of $\mathcal{X}$, indexed by $\mathcal{I}$, with $|\mathcal{I}| = \tilde{m} = [\delta m]$, where $\delta \in (0, 1]$ denotes the sampling rate. With this sampled index set $\mathcal{I}$, we construct matrices $\tilde{Y}, \tilde{Z} \in \mathbb{R}^{n \times m}$, where the $(i, j)$-th entries are defined as follows:

$$\tilde{Y}_{ij} = \frac{P_{ij'}(1 - 2P_{ij'})}{x_i^*}, \quad \tilde{Z}_{ij} = \frac{P_{ij'}}{x_i^*}, \tag{13}$$

with $j'$ being uniformly sampled from $\mathcal{I}$. Given that $\tilde{Y}, \tilde{Z}$ now possess reduced ranks of $[\delta m]$, the computational complexity of evaluating $(D + \tilde{Y}\tilde{Z}^\top)^{-1}$ is significantly lowered to $O(n^2 \tilde{m} + n \tilde{m}^2 + \tilde{m}^3)$. Algorithm 1 describes the steps for addressing OP (7).

---

**Algorithm 1:** Approximated Gradient Descent for Solving OP (7).

**Input:** The environment specified by $\mathcal{G}$, maximum iteration number $T$, sample rate $\delta$, learning rate $\eta$, initial mechanism PPM($\boldsymbol{\beta}$).

1 **for** $t \in [T]$ **do**
2      Find the PNE $\boldsymbol{x}^*$ of $\mathcal{G}$ under PPM($\boldsymbol{\beta}$) using Algorithm 2,
3      Uniformly sample $[\delta m]$ users from $\mathcal{X}$ and use them to compute matrices $\tilde{Y}, \tilde{Z}$ in Eq. (13),
4      Compute the approximated gradient $\frac{dW_\lambda}{d\boldsymbol{\beta}}$ using (8),(9),(12) with sketched matrices $\tilde{Y}, \tilde{Z}$,
5      Update $\boldsymbol{\beta} = \boldsymbol{\beta} + \eta \frac{dW_\lambda}{d\boldsymbol{\beta}}$.

---

Algorithm 1 requires solving for the PNE of $\mathcal{G}$ each time when $\boldsymbol{\beta}$ is updated. To accomplish this, we employ the multi-agent mirror descent method, as proposed in [7] and detailed in Algorithm 2 in Appendix, to serve as a subroutine[5]. To accelerate the convergence of Algorithm 1, the PNE strategy $\boldsymbol{x}^*$ obtained under the previous $\boldsymbol{\beta}$ is used as the initial strategy for computing the new PNE after updating $\boldsymbol{\beta}$. Further implementation details are provided in the experiment section.

## 7 Experiments

To validate our theoretical findings and demonstrate the performance of Algorithm 1, we conduct simulations on instances of $\mathcal{G}$ constructed from both synthetic data and the MovieLens-1m dataset [28]. In our experiments, Algorithm 2 is employed to solve the PNE for each instance of $\mathcal{G}$. Below, we first outline the specifications of these two simulation environments and then present our results.

**Synthetic environment** For the synthetic environment, we construct the user population $\mathcal{X}$ by setting an embedding dimension $d = 32$ and independently sampling 50 cluster centers, denoted as $\{\mathbf{c}_1, \ldots, \mathbf{c}_{50}\}$, from the unit sphere $\mathbb{S}^{d-1}$. For each center $\mathbf{c}_i$, users belonging to cluster-$i$ are generated by sampling independently from a Gaussian distribution $\mathcal{N}(\mathbf{c}_i, 0.5^2 \mathbf{I}_d)$. The sizes of the 50 user clusters are determined uniformly at random, ensuring the total size of $\mathcal{X}$ is $m = 1000$. Similarly, $n = 200$ creators are generated, and the relevance matrix $M \in \mathbb{R}^{n \times m}$ is defined by the dot product between each user-creator pair, which are then normalized to the range $[0, 1]$. This synthetic

---

[5]In [7], the algorithm is guaranteed to converge to PNE under zeroth order feedback. Here we use the perfect gradient as input and thus the convergence is also guaranteed.

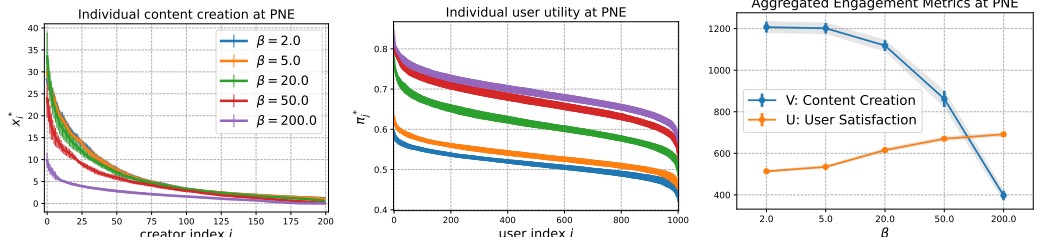

Figure 1: The left and the middle panel: the empirical distributions of content creation frequency $x_i^*$ and each user's individual utility $\pi_j^*$. Different colors represent results for PNEs induced by different $\beta$. Right: the total content creation $V$ and total user satisfaction $U$ obtained under different $\beta$. Error bars obtained from 10 independently generated environments.

dataset encapsulates a class of clustered user and creator preference distributions. On the creators' side, their cost functions are set to $c_i(x) = c_i x^\rho$, with the default $\rho = 1.5$. The marginal costs $\{c_i\}_{i=1}^n$ are randomly sampled from a uniform distribution $\mathcal{U}[0.1, 0.5]$.

**Environment constructed from MovieLens-1m dataset** We use deep matrix factorization [22] to train user and movie embeddings (with dimension set to 32) by fitting the observed ratings in the range of 1 to 5. To ensure the quality of the trained embeddings, we performed a 5-fold cross-validation and obtained an averaged RMSE=0.739 on the test sets. With the same hyper-parameter, we train the user/item embeddings with the complete dataset. We randomly select $m = 1000$ user embeddings to construct the population $\mathcal{X}$ and $n = 200$ movie embeddings as the creator profiles. Similarly, The relevance matrix $M \in \mathbb{R}^{n \times m}$ is given by the dot product between each user-creator pair normalized to $[0, 1]$ and creators' cost functions are the same as we specified in the synthetic environment.

### 7.1 The Empirical Trade-Offs Between $U$ and $V$

Figure 1 illustrates the content creation frequency $x_i^*$, user utility $\pi_j^*$, and their corresponding aggregated values $U = \sum_j \pi_j^*$, $V = \sum_i x_i^*$ under the PNE induced by different homogeneous $\beta$ (i.e., all users share the same $\beta$). The result in the right panel shows that a larger $\beta$ enhances overall user satisfaction $U$ but undermines total content creation $V$. As $\beta$ increases, the drop in $V$ becomes more significant. This empirical finding supports Theorem 2 and suggests it holds under broader settings without the assumptions on creator cost function and user population structure. The left and middle plots illustrate each creator $i$'s creation frequency $x_i^*$ and each user $j$'s utility $\pi_j^*$ at the PNE, such that both $x_i^*$ and $\pi_j^*$ are rearranged in descending order. They show that when $\beta$ is shared across all users, its change affects $x_i^*$ and $\pi_j^*$ in the same direction.

### 7.2 The Optimal PPM($\beta$) Found by Algorithm 1

Next, we use Algorithm 1 to find the optimal PPM($\boldsymbol{\beta}$) and investigate the properties of the optimal $\boldsymbol{\beta}$. We set $\lambda = 0.5$ and aim to maximize the objective $W_\lambda = U + 0.5V$, more results under different choices of $\lambda$ can be found in Appendix B. The initial $\boldsymbol{\beta}$ is set to $(100, \cdots, 100)$, representing a nearly deterministic matching for every user. Algorithm 1 is then run to update $\boldsymbol{\beta}$. The sample rate and learning rate are set to $\delta = 0.1, \eta = 200$. In addition to searching for personalized $\beta_j$ for each user $j$, we also attempt to find a homogeneous $\beta$ (i.e., a fixed $\beta_j$ for each $j$) using Algorithm 1[6].

The first and third panels in Figure 2 show the evolution of $W_\lambda$ during the optimization process in both synthetic and MovieLens environments. As illustrated, Algorithm 1 successfully finds a better PPM($\boldsymbol{\beta}$) compared to the baseline of exact matching for all users, with a significant gain of over 20% in the welfare metric. Furthermore, in both environments, personalized $\beta$ leads to a slightly better outcome compared to homogeneous $\beta$.

The second and fourth panels depict the optimal $\beta_j$ for each user $j$, arranged in descending order. These panels provide insights into how such a mechanism achieves better trade-offs. For each user

---

[6]The gradient of $W_\lambda$ with respect to a homogeneous $\beta$ can be readily obtained by summing all the partial derivatives of $W_\lambda$ with respect to $\beta_j$.

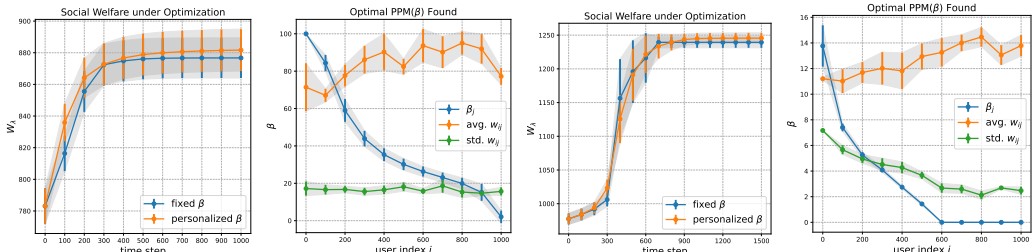

Figure 2: Panel 1,2: social welfare improving curve under Algorithm 1, and the distribution of the obtained optimal $\beta_j$ in the synthetic environment. Panel 3,4: the same plots in the MovieLens environment. $\lambda = 0.5$.

index $j$ on the $x$-axis, we also plot the average and the standard deviation of the relevance scores $\{w_{ij}\}_{i=1}^n$ associated with each user $j$ over all creators, shown as orange and green lines. Based on the definition, users with smaller average scores and higher standard deviations are considered more "picky" or selective, indicating high relevance scores with a small group of creators and low scores with many others. Conversely, users with higher average scores and smaller standard deviations are less selective and more open to exploration. The results show that the optimal $\boldsymbol{\beta}$ tends to increase the exploration strengths (by deploying smaller $\beta_j$) for less selective users. This approach is intuitive, as it safely increases exploration while minimizing losses in user engagement.

## 8 Conclusion

In this work, we introduced a new game-theoretical model $C^4$ (Cournot Content Creation Competition) to explore how creators strategically determine their creation frequency under a UGC platform's recommendation algorithm. Our investigations reveal a critical balance between user satisfaction and creator engagement, mediated by the exploration strength of the recommendation. The existence and uniqueness of the PNE of $C^4$ games provide a predictive framework for assessing the effects of algorithmic choices on content diversity and volume. Through both theoretical analysis and empirical simulations, we demonstrated how varying the exploration strength can either enhance user engagement at the cost of reduced content diversity or encourage richer content creation at the expense of immediate user satisfaction. These findings disclose the delicate trade-offs platform designers face and highlight the utility of our model as a pre-deployment audit tool for optimizing recommendation algorithms to balance platforms' long-term and short-term objectives.

While our $C^4$ model offers insights into strategic differentiation among creators regarding production quantity, it relies on a simplified assumption that creators maintain a fixed niche, consistently producing content on the same topic with similar quality. This assumption, though useful for modeling purposes, may be restrictive in real-world scenarios where creators dynamically adjust topics, vary content quality, and scale production quantity. Exploring the dynamics where creators compete across heterogeneous dimensions—such as topic variety, content quality, and production quantity—would be a valuable direction for future research. We leave this intriguing problem for future work.

**Acknowledgment.** This work is supported in part by the NSF Award IIS-2128019, NSF Award CCF-2303372, AI2050 program at Schmidt Sciences (Grant G-24-66104) and Army Research Office Award W911NF-23-1-0030.

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

# Appendix to *Unveiling User Satisfaction and Creator Productivity Trade-Offs in Recommendation Platforms*

## A  Omitted Proofs

### A.1  Proof of Theorem 1

*Proof.* First of all, we argue that given any $\mathcal{G}(n, m, M, \{c_i\}, \boldsymbol{\beta})$, for any creator $i$, there exists an $\delta_i > 0$ such that any $x_i \in [0, \delta_i]$ cannot be an equilibrium strategy. This is because given any $\boldsymbol{x}_{-i} \in \mathbb{R}_{\geq 0}^{n-1}$, $u_i$ as a function of $x_i$ has a continuous and strictly positive gradient at $x_i = 0$, meaning that there exists a $\delta_i > 0$ such that $\frac{\partial u_i}{\partial x_i}\Big|_{x_i=t} > 0, \forall t \in [0, \delta_i]$ regardless of what other creators' strategies are. In other word, for any $x_i \leq \delta_i$, creator $i$ can always increase her strategy to strictly improve her utility. As a result, any potential PNE $\boldsymbol{x}^*$ must satisfy that $x_i^* \geq \delta_i$.

On the other hand, since $c_i(x_i) \to +\infty$ when $x_i \to +\infty$ but the traffic gain for each creator is at most $m$, we have $u_i(x_i, \boldsymbol{x}_{-i}) \to -\infty, \forall \boldsymbol{x}_{-i} \in \mathbb{R}_{\geq 0}^{n-1}$ when $x_i \to +\infty$. As a result, any equilibrium strategy must also be upper bounded by a uniform constant $\Delta > 0$.

To argue the existence and uniqueness of PNE of $\mathcal{G}$, in the following we may with out loss of generality restrict each creator $i$'s strategy set to a convex set $[\delta_i, \Delta]$.

For any fixed $\boldsymbol{\beta} = (\beta_1, \cdots, \beta_m)$, let $a_{ij} = \exp(\beta_j w_{ij})$ and Eq. (1) can be simplified to

$$u_i(x_i, \boldsymbol{x}_{-i}; \boldsymbol{\beta}) = \sum_{j=1}^{m} \left( \frac{x_i a_{ij}}{\sum_{k=1}^{n} x_k a_{kj}} \right) - c_i(x_i), \tag{14}$$

For simplicity we denote $g_j(\boldsymbol{x}) = \sum_{k=1}^{n} x_k a_{kj}$ and $u_i$ can be expressed as $u_i(x_i, \boldsymbol{x}_{-i}) = \sum_{j=1}^{m} \frac{x_i a_{ij}}{g_j(\boldsymbol{x})} - c_i(x_i)$. Our proof starts from a sufficient condition from [44] for a game to be monotone. A game is said to satisfy the *diagonal strict concavity* (DSC) condition if (1) each player has a concave utility function in his own strategy in a convex strategy space; and (2) there exists *some* non-zero parameter $\lambda = (\lambda_1, \cdots, \lambda_n)$ such that the Hessian matrix given by

$$H_{kl}(\boldsymbol{x}; \lambda) \triangleq \frac{\lambda_k}{2} \frac{\partial^2 u_k(\boldsymbol{x})}{\partial x_k \partial x_l} + \frac{\lambda_l}{2} \frac{\partial^2 u_l(\boldsymbol{x})}{\partial x_l \partial x_k} \tag{15}$$

is *strictly* negative-definite. In [44], it is shown that any game satisfying $\lambda$-DSC condition has a unique pure Nash equilibrium (PNE); such a game is often referred to as monotone games.

First of all, we already argued that each creator $i$'s strategy set is $[\delta_i, \Delta]$, which is a convex set. Core to our proof is to show that game $\mathcal{G}$ is 1-DSC under the theorem conditions. Direct calculation shows that for any $1 \leq k \leq l \leq n$,

$$\frac{\partial^2 u_k(\boldsymbol{x})}{\partial x_k \partial x_l} = \frac{a_{kj} a_{lj}}{g_j^3} \cdot (-g_j + 2a_{kj} x_k),$$

$$\frac{\partial^2 u_l(\boldsymbol{x})}{\partial x_l \partial x_k} = \frac{a_{kj} a_{lj}}{g_j^3} \cdot (-g_j + 2a_{lj} x_l),$$

and therefore the Hessian matrix of $\mathcal{G}$ specified by the RHS of Eq. (15) is equal to

$$
-[H(\boldsymbol{x})] = \sum_{j=1}^{m} g_j^{-3} \begin{bmatrix} a_{1j} \\ a_{2j} \\ \vdots \end{bmatrix} \begin{bmatrix} 2\sum_{i \neq 1} x_i a_{ij} & \sum_{i \notin \{1,2\}} x_i a_{ij} & \cdots \\ \sum_{i \notin \{1,2\}} x_i a_{ij} & 2\sum_{i \neq 2} x_i a_{ij} & \cdots \\ \vdots & \vdots & \ddots \end{bmatrix} [a_{1j}, a_{2j}, \ldots] + \begin{bmatrix} \frac{\partial^2 c_1}{\partial x_1^2} & 0 & \cdots \\ 0 & \frac{\partial^2 c_2}{\partial x_2^2} & \cdots \\ \vdots & \vdots & \ddots \end{bmatrix}
$$

$$\triangleq \sum_{j=1}^{m} g_j^{-3} \boldsymbol{a}_j H_j \boldsymbol{a}_j^\top + H_0, \tag{16}$$

where $g_j$ in the above expressions denotes $g_j(\boldsymbol{x})$, vector $\boldsymbol{a}_j = (a_{1j}, \cdots, a_{nj})^\top$. We can see that if all the cost functions are strictly convex, the second diagonal matrix $H_0$ in the RHS of Eq. (16) is strictly positive-definite (PD). Therefore, it suffices to show that (1) for all $j \in [m]$, $H_j$ is PD. To see this, let $z_i = x_i a_{ij}$ and we show that for any $\boldsymbol{y} = (y_1, \cdots, y_n) \in \mathbb{R}^n$, $\boldsymbol{y}^\top H_j \boldsymbol{y} \geq 0$, and the equality holds if and only if $\boldsymbol{y} = \boldsymbol{0}$. In fact, note that

$$
\begin{aligned}
\boldsymbol{y}^\top H_j \boldsymbol{y} &= 2 \sum_{i=1}^n y_i^2 \left( \sum_{j \neq i} z_j \right) + 2 \sum_{i<j} y_i y_j \left( \sum_{k \notin \{i,j\}} z_k \right) \\
&= \sum_{i=1}^n y_i^2 \left( \sum_{j \neq i} z_j \right) + \left[ \sum_{i=1}^n y_i^2 \left( \sum_{j \neq i} z_j \right) + 2 \sum_{i<j} y_i y_j \left( \sum_{k \notin \{i,j\}} z_k \right) \right] \\
&= \sum_{i=1}^n y_i^2 \left( \sum_{j \neq i} z_j \right) + \sum_{k=1}^n z_k \left[ \sum_{j \neq k} y_j^2 + 2 \sum_{i<j, i \neq k, j \neq k} y_i y_j \right] \\
&= \sum_{i=1}^n y_i^2 \left( \sum_{j \neq i} z_j \right) + \sum_{i=1}^n z_i \left( \sum_{j \neq i} y_j \right)^2 \geq 0.
\end{aligned}
\tag{17}
$$

Because $x_i$ and $a_i$ are all strictly positive, each $z_i$ must also be strictly positive. Hence, Eq. (17) can take value zero if and only if $y_i = 0, \forall i \in [n]$. Therefore, $H_j$ is PD for any $j \in [m]$, which completes the proof.

$\square$

## A.2   Proof of Theorem 2

First let's recall the definition of $U, V$ when $m = 1$:

$$
U(\boldsymbol{x}^*(\beta); \beta) = \sum_{i=1}^n \left( \frac{w_i x_i^* e^{\beta w_i}}{\sum_{k=1}^n x_k^* e^{\beta w_k}} \right),
\tag{18}
$$

$$
V(\boldsymbol{x}^*) = \sum_{i=1}^n x_i^*.
\tag{19}
$$

In the following, we prove the monotonicity of $U(\beta)$ and $V(\beta)$ by showing $\frac{d \ln U}{d\beta} > 0$ and $\frac{dV}{d\beta} < 0$, respectively. Before presenting the detailed proof, we first derive some relevant definitions and their properties that will be used in the proof.

For simplicity we omit the superscript $^*$ in $\boldsymbol{x}^*$ and simply use $\boldsymbol{x}$ to refer to the PNE of $\mathcal{G}$. When $m = 1$, the creator utility function writes

$$
u_i(x_i, x_{-i}) = \frac{x_i e^{\beta w_i}}{\sum_{k=1}^n x_k e^{\beta w_k}} - c_i(x_i), i \in [n].
\tag{20}
$$

Let $P_i = \frac{x_i a_i}{\sum_{k=1}^n x_k a_k}$, where $a_i = e^{\beta w_i}$. First of all, we claim that it is without loss of generality to consider the regime where $P_i \leq \frac{1}{3}$. To see this, consider the following two $C^4$ instances:

$$
\begin{aligned}
&\mathcal{G}_1(n, m = 1, \boldsymbol{w} = (w_1, \cdots, w_n), \mathbf{c} = (c_1, \cdots, c_n), \boldsymbol{\beta}), \\
&\mathcal{G}_2(3n, m = 1, [\boldsymbol{w}, \boldsymbol{w}, \boldsymbol{w}], [\mathbf{c}/3, \mathbf{c}/3, \mathbf{c}/3], [\boldsymbol{\beta}, \boldsymbol{\beta}, \boldsymbol{\beta}]).
\end{aligned}
$$

Clearly, both games $\mathcal{G}_1, \mathcal{G}_2$ have unique PNE. Let the PNE of $\mathcal{G}_1$ be denoted by $\boldsymbol{x}_1^*$. In $\mathcal{G}_2$, since its $3n$ players are divided into three identical groups, its PNE can be represented as $(\boldsymbol{x}_2^*, \boldsymbol{x}_2^*, \boldsymbol{x}_2^*)$, where each group of $n$ players follow the same strategy. Moreover, for any $1 \leq i \leq n$, it is straightforward to observe that the $i$-th player's utility functions in $\mathcal{G}_1$ and $\mathcal{G}_2$ differ only by a multiplicative constant of 3. Consequently, we have $\boldsymbol{x}_1^* = \boldsymbol{x}_2^*$. Therefore, for any $C^4$ instance $\mathcal{G}_1$ with $m = 1$, we can always

construct an equivalent instance $\mathcal{G}_2$ that shares the same PNE structure, while ensuring $P_i \leq \frac{1}{3}$. This justifies the assumptions that, without loss of generality, we can take $P_i \leq \frac{1}{3}$.

Another property we need is that the PNE strategy $x_i$ of any player $i$ in $\mathcal{G}_1$ is bounded in a compact region $[0, L]$ for some constant $L$, regardless of the values of $\boldsymbol{\beta}$. To see this, note that $c_i(x_i)$ is increasing in $x_i$ and goes to infinity as $x_i \to +\infty$ while $P_i$ is upper bounded by 1. As a result, for any $x_i$ such that $u_i(x_i, \boldsymbol{x}_{-i}) \leq 1 - c_i(x_i) < -c_i(0) = u_i(0, \boldsymbol{x}_{-i})$, $x_i$ cannot be a PNE strategy as switching to 0 increases player $i$'s utility. Therefore, if we take

$$L = \max_{i \in [n]} \{ \inf_x \{ x \geq 0 : 1 - c_i(x_i) < -c_i(0) \} \},$$

it holds that $x_i \leq L, \forall i \in [n]$.

Let $F(x, \beta) = \left( \frac{\partial u_i}{\partial x_i} \right)_{i=1}^n$ be an $n$-value function. From Corollary 1 we know $\boldsymbol{x}, \beta$ satisfy $F(\boldsymbol{x}, \beta) = 0$. And Theorem 1 guarantees that for any $\beta \geq 0$, the $\boldsymbol{x}$ implicitly determined by $F(\boldsymbol{x}, \beta) = 0$ exists and is unique. Therefore, by the implicit function theorem [37], the derivative of $\boldsymbol{x}$ w.r.t. $\beta$ can be written as

$$\frac{d\boldsymbol{x}}{d\beta} = -\left( \frac{\partial F}{\partial \boldsymbol{x}} \right)^{-1} \cdot \frac{\partial F}{\partial \beta},$$

where $\left[ \frac{\partial F}{\partial \boldsymbol{x}} \right]_{n \times n}$ is the Jacobian matrix (which is also the Hessian of $\mathcal{G}$ when $\lambda = 1$, see Eq. (15)) and $\frac{\partial F}{\partial \beta} \in \mathbb{R}^{n \times 1}$ is the partial derivative of $F$ w.r.t. $\beta$.

The first-order derivative of $u_i$ can be calculated as

$$\frac{\partial u_i}{\partial x_i} = \frac{e^{\beta w_i}}{\sum_{k=1}^n x_k e^{\beta w_k}} - x_i \left( \frac{e^{\beta w_i}}{\sum_{k=1}^n x_k e^{\beta w_k}} \right)^2 - c_i'(x_i), \tag{21}$$

and we can use it to further obtain the following explicit expressions in terms of the derivatives of $F$:

$$\left( \frac{\partial F_i}{\partial x_i} \right) = -\frac{1}{x_i^2} P_i^2 (1 - 2P_i) - \frac{1}{x_i^2} P_i^2 - c_i''(x_i),$$

$$\left( \frac{\partial F_i}{\partial x_j} \right) = -\frac{1}{x_i x_j} P_i P_j (1 - 2P_i), j \neq i,$$

$$\left( \frac{\partial F_i}{\partial \beta} \right) = \frac{P_i}{x_i} \cdot (1 - 2P_i) \cdot \left( w_i - \sum_{k=1}^n P_k w_k \right). \tag{22}$$

Let's define a positive definite diagonal matrix

$$D = \text{diag} \left( \frac{P_1^2}{x_1^2} + c_1'', \cdots, \frac{P_n^2}{x_n^2} + c_n'' \right).$$

Since $c_i'' > 0$, we can introduce variables

$$\delta_i \in (0, 1) \text{ such that } \frac{P_i^2}{x_i^2} + c_i'' = \frac{P_i^2}{\delta_i x_i^2}. \tag{23}$$

In addition, let's also define

$$\boldsymbol{y} = \left( \frac{P_1(1 - 2P_1)}{x_1}, \cdots, \frac{P_n(1 - 2P_n)}{x_n} \right)^\top, \boldsymbol{z} = \left( \frac{P_1}{x_1}, \cdots, \frac{P_n}{x_n} \right)^\top, \tag{24}$$

then we have $\frac{\partial F}{\partial \boldsymbol{x}} = D + \boldsymbol{y} \boldsymbol{z}^\top$ and from Sherman–Morrison formula[46], it holds that

$$-\left( \frac{\partial F}{\partial \boldsymbol{x}} \right)^{-1} = (D + \boldsymbol{y} \boldsymbol{z}^\top)^{-1}$$

$$= D^{-1} - \frac{D^{-1} \boldsymbol{y} \boldsymbol{z}^\top D^{-1}}{1 + \boldsymbol{z}^\top D^{-1} \boldsymbol{y}}, \tag{25}$$

where

$$D^{-1} = \mathrm{diag}\left(\frac{\delta_1 x_1^2}{P_1^2}, \cdots, \frac{\delta_n x_n^2}{P_n^2}\right),$$

$$D^{-1}\boldsymbol{y} = \left(\frac{\delta_1 x_1(1-2P_1)}{P_1}, \cdots, \frac{\delta_n x_n(1-2P_n)}{P_n}\right)^\top,$$

$$\boldsymbol{z}^\top D^{-1} = \left(\frac{\delta_1 x_1}{P_1}, \cdots, \frac{\delta_n x_n}{P_n}\right).$$

Since $P_i \leq \frac{1}{3}$, we have $1 - 2P_i \geq \frac{1}{3} > 0$. With all the notations introduced so far we are now ready to give the formal proof of Theorem 2.

*Proof.* We prove the monotonicity of $U(\beta)$ and $V(\beta)$ by showing $\frac{d\ln U}{d\beta} > 0$ and $\frac{dV}{d\beta} < 0$.

**The monotonicity of $U(\beta)$:** The first-order derivative of $\ln U$ w.r.t. $\beta$ is given by

$$\begin{aligned}
\frac{d\ln U}{d\beta} &= \frac{1}{U} \cdot \left(\frac{\partial U}{\partial \boldsymbol{x}} \cdot \frac{d\boldsymbol{x}}{d\beta} + \frac{\partial U}{\partial \beta}\right) \\
&= -\frac{1}{U} \cdot \frac{\partial U}{\partial \boldsymbol{x}} \cdot \left(\frac{\partial F}{\partial \boldsymbol{x}}\right)^{-1} \cdot \frac{\partial F}{\partial \beta} + \frac{1}{U} \cdot \frac{\partial U}{\partial \beta}.
\end{aligned} \tag{26}$$

where $\frac{\partial U}{\partial \boldsymbol{x}} \in \mathbb{R}^{1\times n}$ is the partial derivative of $U$ w.r.t. $\boldsymbol{x}$. Let $a_i = e^{\beta w_i}$, we will first show $\frac{1}{U} \cdot \frac{\partial U}{\partial \beta} \geq 0$. In fact, calculation shows

$$\begin{aligned}
\frac{1}{U} \cdot \frac{\partial U}{\partial \beta} &= \frac{\sum_{k=1}^n x_k a_k}{\sum_{k=1}^n w_k x_k a_k} \cdot \left(\frac{(\sum_{k=1}^n w_k^2 x_k a_k)(\sum_{k=1}^n x_k a_k) - (\sum_{k=1}^n w_k x_k a_k)^2}{(\sum_{k=1}^n x_k a_k)^2}\right) \\
&= \frac{\sum_{k=1}^n w_k^2 x_k a_k}{\sum_{k=1}^n w_k x_k a_k} - \frac{\sum_{k=1}^n w_k x_k a_k}{\sum_{k=1}^n x_k a_k}.
\end{aligned} \tag{27}$$

From Cauchy–Schwarz inequality, it holds that

$$\sum_{k=1}^n w_k^2 x_k a_k \cdot \sum_{k=1}^n x_k a_k \geq \left(\sum_{k=1}^n \sqrt{w_k^2 x_k a_k \cdot x_k a_k}\right)^2 = \left(\sum_{k=1}^n w_k x_k a_k\right)^2.$$

Therefore, the RHS of Eq. (27) is greater than or equal to 0. Hence, it suffices to show

$$-\frac{1}{U} \cdot \frac{\partial U}{\partial \boldsymbol{x}} \cdot \left(\frac{\partial F}{\partial \boldsymbol{x}}\right)^{-1} \cdot \frac{\partial F}{\partial \beta} > 0. \tag{28}$$

Also note that

$$\begin{aligned}
\frac{1}{U} \cdot \frac{\partial U}{\partial x_i} &= \frac{\sum_{k=1}^n x_k a_k}{\sum_{k=1}^n w_k x_k a_k} \cdot \left(\frac{w_i a_i(\sum_{k=1}^n x_k a_k) - a_i \sum_{k=1}^n w_k x_k a_k}{(\sum_{k=1}^n x_k a_k)^2}\right) \\
&= \frac{w_i a_i}{\sum_{k=1}^n w_k x_k a_k} - \frac{a_i}{\sum_{k=1}^n x_k a_k} \\
&= \frac{w_i a_i}{\sum_{k=1}^n w_k x_k a_k} - \frac{P_i}{x_i},
\end{aligned} \tag{29}$$

and substitute Eq. (22), (25), and (29) into the LHS of Eq. (28), we obtain

$$\begin{aligned}
&-\frac{1}{U} \cdot \frac{\partial U}{\partial \boldsymbol{x}} \cdot \left(\frac{\partial F}{\partial \boldsymbol{x}}\right)^{-1} \cdot \frac{\partial F}{\partial \beta} \\
&= \left[\frac{w_i a_i}{\sum_{k=1}^n w_k x_k a_k} - \frac{P_i}{x_i}\right]_{i\in[n]}^\top \cdot \left[D^{-1} - \frac{D^{-1}\boldsymbol{y}\boldsymbol{z}^\top D^{-1}}{1 + \boldsymbol{z}^\top D^{-1}\boldsymbol{y}}\right] \cdot \left[\frac{P_i}{x_i} \cdot (1-2P_i) \cdot \left(w_i - \sum_{k=1}^n P_k w_k\right)\right]_{i\in[n]} \\
&= \frac{1}{T}\sum_{i=1}^n \delta_i(1-2P_i)(w_i - T)^2 - \frac{(\sum_{i=1}^n \delta_i(1-2P_i)(w_i - T))^2}{T(1 + \sum_{i=1}^n \delta_i(1-2P_i))},
\end{aligned} \tag{30}$$

where $T = \frac{\sum_{k=1}^n w_k x_k a_k}{\sum_{k=1}^n x_k a_k} = \sum_{i=1}^n P_i w_i$. Therefore, it suffices to prove

$$\sum_{i=1}^n \delta_i (1 - 2P_i)(w_i - T)^2 - \frac{\left(\sum_{i=1}^n \delta_i (1 - 2P_i)(w_i - T)\right)^2}{1 + \sum_{i=1}^n \delta_i (1 - 2P_i)} > 0. \tag{31}$$

Since $1 - 2P_i > 0$, from Cauchy–Schwarz inequality it holds that

$$\sum_{i=1}^n \delta_i (1 - 2P_i)(w_i - T)^2 \cdot \left(1 + \sum_{i=1}^n \delta_i (1 - 2P_i)\right)$$

$$= \sum_{i=1}^n \delta_i (1 - 2P_i)(w_i - T)^2 \cdot \sum_{i=1}^n \delta_i (1 - 2P_i) + \sum_{i=1}^n \delta_i (1 - 2P_i)(w_i - T)^2$$

$$\geq \left(\sum_{i=1}^n \delta_i (1 - 2P_i)(w_i - T)\right)^2 + \sum_{i=1}^n \delta_i (1 - 2P_i)(w_i - T)^2$$

$$> \left(\sum_{i=1}^n \delta_i (1 - 2P_i)(w_i - T)\right)^2,$$

where the last inequality holds because $\{w_i\}$ are not identical so there exists at least one $j \in [n]$ such that $\delta_j (1 - 2P_j)(w_j - T)^2 > 0$. Therefore, Eq. (31) holds and we have $\frac{d \ln U}{d\beta} > 0$.

**The monotonicity of $V(\beta)$:** Next we show $\frac{dV}{d\beta} < 0$. Since $\frac{\partial V}{\partial \beta} = 0$, this is equivalent to show

$$\frac{\partial V}{\partial \boldsymbol{x}} \cdot \left(\frac{\partial F}{\partial \boldsymbol{x}}\right)^{-1} \cdot \frac{\partial F}{\partial \beta} > 0. \tag{32}$$

Note that $\frac{dV}{dx_i} = 1$, we have

$$\frac{\partial V}{\partial \boldsymbol{x}} \cdot \left(\frac{\partial F}{\partial \boldsymbol{x}}\right)^{-1} \cdot \frac{\partial F}{\partial \beta}$$

$$= - [1, 1, \cdots, 1] \cdot \left[D^{-1} - \frac{D^{-1} \boldsymbol{y} \boldsymbol{z}^\top D^{-1}}{1 + \boldsymbol{z}^\top D^{-1} \boldsymbol{y}}\right] \cdot \left[\frac{P_i}{x_i} \cdot (1 - 2P_i) \cdot \left(w_i - \sum_{k=1}^n P_k w_k\right)\right]_{i \in [n]}$$

$$= - \sum_{i=1}^n \frac{\delta_i x_i (1 - 2P_i)(w_i - T)}{P_i} + \frac{1}{1 + \sum_{i=1}^n \delta_i (1 - 2P_i)} \sum_{i=1}^n \frac{\delta_i x_i (1 - 2P_i)}{P_i} \sum_{i=1}^n \delta_i (1 - 2P_i)(w_i - T). \tag{33}$$

To show the RHS of Eq. (33) is positive, it suffices to show

$$\left(1 + \sum_{i=1}^n \delta_i (1 - 2P_i)\right) \sum_{i=1}^n \frac{\delta_i x_i (1 - 2P_i)(w_i - T)}{P_i} < \sum_{i=1}^n \frac{\delta_i x_i (1 - 2P_i)}{P_i} \sum_{i=1}^n \delta_i (1 - 2P_i)(w_i - T). \tag{34}$$

Plugin $T = \sum_{i=1}^n P_i w_i$ into Eq. (34), it is equivalent to show that

$$\left(1 + \sum_{i=1}^n \delta_i (1 - 2P_i)\right) \sum_{i=1}^n \frac{\delta_i x_i (1 - 2P_i) w_i}{P_i} - \sum_{i=1}^n P_i w_i \sum_{i=1}^n \frac{\delta_i x_i (1 - 2P_i)}{P_i} < \sum_{i=1}^n \frac{\delta_i x_i (1 - 2P_i)}{P_i} \sum_{i=1}^n \delta_i (1 - 2P_i) w_i,$$

and note that $1 = \sum_{i=1}^n P_i$, it is equivalent to show

$$\sum_{i=1}^n [\delta_i (1 - 2P_i) + P_i] \sum_{i=1}^n \frac{\delta_i x_i (1 - 2P_i) w_i}{P_i} < \sum_{i=1}^n \frac{\delta_i x_i (1 - 2P_i)}{P_i} \sum_{i=1}^n [\delta_i (1 - 2P_i) + P_i] w_i. \tag{35}$$

Let $a_i = \frac{\delta_i x_i (1 - 2P_i)}{P_i}, b_i = \delta_i (1 - 2P_i) + P_i$, then Eq. (35) is equivalent to

$$\sum_{i=1}^{n} b_i \cdot \sum_{i=1}^{n} a_i w_i < \sum_{i=1}^{n} a_i \cdot \sum_{i=1}^{n} b_i w_i \iff \sum_{i>j} (w_i - w_j) \left( \frac{b_i}{a_i} - \frac{b_j}{a_j} \right) > 0.$$

Next, without loss of generality we show that for any $1 \le i < j \le n$, if $w_i > w_j$ then it also holds that $\frac{b_i}{a_i} > \frac{b_j}{a_j}$ for sufficiently large $\beta$. In fact, from $P_i = \frac{x_i e^{\beta w_i}}{\sum_{k=1}^{n} x_k e^{\beta w_k}}$ and Eq. (23) we obtain

$$
\begin{aligned}
\frac{b_i}{a_i} \cdot \frac{a_j}{b_j} &= \frac{\delta_j (\delta_i (1 - 2P_i) + P_i)(1 - 2P_j)}{\delta_i (\delta_j (1 - 2P_j) + P_j)(1 - 2P_i)} \cdot e^{\beta(w_i - w_j)} \\
&= \frac{1 + P_i \delta_i / (1 - 2P_i)}{1 + P_j \delta_j / (1 - 2P_j)} \cdot e^{\beta(w_i - w_j)} \\
&= \frac{1 + \frac{P_i^3}{(1 - 2P_i)(P_i^2 + x_i^2 c_i''(x_i))}}{1 + \frac{P_j^3}{(1 - 2P_j)(P_j^2 + x_j^2 c_j''(x_j))}} \cdot e^{\beta(w_i - w_j)}.
\end{aligned}
\tag{36}
$$

On the one hand, because $1 - 2P_i, 1 - 2P_j \in [\frac{1}{3}, 1]$, $x_i, x_j \in [0, L]$, and $c_i''(x_i), c_j''(x_j) > 0$, the first terms of the LHS of Eq. (36) is a positive number lower bounded away from zero. On the other hand, $w_i > w_j$ ensures that the second term $e^{\beta(w_i - w_j)}$ can be arbitrarily large as long as $\beta$ is sufficiently large. Therefore, there must exist a $\beta_0 > 0$ such that for any $\beta > \beta_0$, $\frac{b_i}{a_i} \cdot \frac{a_j}{b_j} > 1$ holds for any $i > j$. As a result, $(w_i - w_j) \left( \frac{b_i}{a_i} - \frac{b_j}{a_j} \right) > 0$ holds, which completes the proof.

$\square$

# B  Additional Experiments

We use the following Multi-agent Mirror Descent (MMD) algorithm as the PNE solver of $C^4$, whose convergence is guaranteed by [7]. Since each creator's strategy set $\mathcal{X}_i = [0, +\infty)$, we can simply choose a projection mapping $\text{Proj}_{\mathcal{X}_i}(\boldsymbol{x}) = (\max(x_i, 0))_{i=1}^{n}$. The gradients of utility functions can be implemented directly since they have closed forms. Through our experiment, the default $T = 10000, \eta = 0.1, \epsilon = 1e - 2, x_i^{(0)} = 1.0$. Algorithm 2 is a simplified version of Algorithm 1 in [7] where we replace the gradient estimation to the exact gradient. According to Theorem 5.1 in [7], Algorithm 2 converges to the unique PNE of any $C^4$ game with probability 1.

---

**Algorithm 2:** Multi-agent Mirror Descent (MMD) with perfect gradient

---

**Input:** Maximum iteration number $T$, step size $\eta$, each player $i$'s utility function $u_i$, error
tolerance $\epsilon$, initial strategy $\boldsymbol{x}_i = \boldsymbol{x}_i^{(0)}$.

1 **for** $t \in [T]$ **do**
2    **if** $\|(\boldsymbol{g}_1, \cdots, \boldsymbol{g}_n)\|_2 < \epsilon$, **then**
3      Break
4    Compute gradient $\boldsymbol{g}_i = \nabla_i u_i(\boldsymbol{x}_i, \boldsymbol{x}_{-i})$, for $i \in [n]$,
5    Update $\boldsymbol{x}_i \leftarrow \text{Proj}_{\mathcal{X}_i}(\boldsymbol{x}_i + \eta \boldsymbol{g}_i), \forall i \in [n]$.

**Output :** $(\boldsymbol{x}_1, \cdots, \boldsymbol{x}_n)$.

---

Figure 3 illustrates the trade-off between $U$ and $V$ in the MovieLens environment, and Figure 4 plots the same information as shown in Figure 2 but with a different value of $\lambda = 0.1$. As we can see, the optimal PPM($\beta$) found by Algorithm 1 conveys a consistent message: it prioritizes the recommendation accuracy for users with more determined preferences while increasing the exploration strength for less selective users.

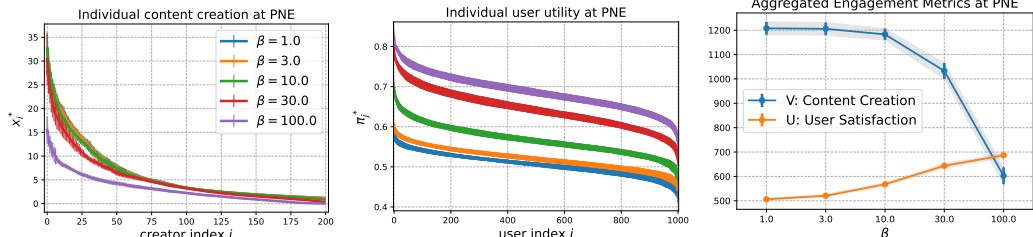

Figure 3: The left and the middle panel: the empirical distributions of content creation frequency $x_i^*$ and each user's individual utility $\pi_j^*$. Different colors represent results for PNEs induced by different $\beta$. Right: the total content creation $V$ and total user satisfaction $U$ obtained under different $\beta$. Error bars obtained from 10 independently generated environments.

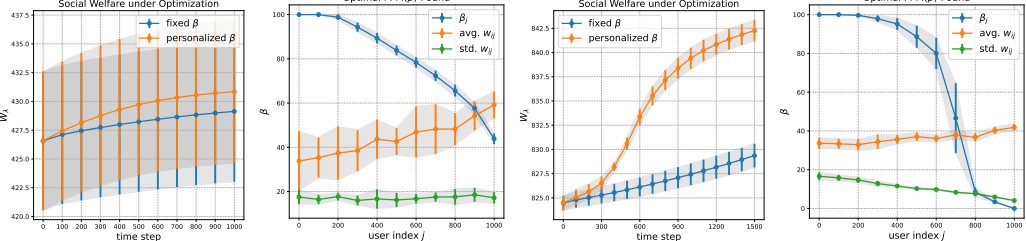

Figure 4: Panel 1,2: social welfare improving curve under Algorithm 1, and the distribution of the obtained optimal $\beta_j$ in the synthetic environment. Panel 3,4: the same plots in the MovieLens environment. $\lambda = 0.1$.

