# OpenReview forum: "Unveiling User Satisfaction and Creator Productivity Trade-Offs in Recommendation Platforms"
_NeurIPS.cc/2024/Conference — NeurIPS 2024 poster_

### Official Review · Reviewer_1Tuv · 2024-07-07

**Soundness:** 3
**Presentation:** 4
**Contribution:** 3
**Rating:** 7
**Confidence:** 4

**Summary:**

This paper studies a model of content creation and consumption on arbitrary online user-generated content platforms (e.g., YouTube, TikTok). It focuses on a type of Cournot competition in which creators mainly modify their creation volume. The paper provides a description of this model, a theoretical analyses of the Pure Nash Equilibria in this setting, an analysis of how platform designers might use mechanism design to balance consumer and creator utility, a framing of this balancing problem as an optimization problem solvable via (approximated) gradient descent, and experiments using purely synthetic data (sampled "users" with Gaussian preferences) and empirical data (users with preferences from the MovieLens dataset, popular in recommender systems).

**Strengths:**

Overall, this paper provides a strong overall contribution and number of results and insights that will be of interest to a number of different communities -- researchers interested in UGC and online communities, mechanism design, ML for social media, etc.

The clarity is high throughout. The paper begins with strong and well argued motivation, the organization is helpful, and in general the overall narrative of the paper is clear.

In terms of novelty, this paper directly builds on a previous modelling work, but is very upfront about highlighting what the main differences and additions are in terms of contribution. The experiments seem to especially build off the design of [40] (esp. in terms of the synthetic data + MovieLens combination), which might be worth mentioning if that is intentional.

Overall, the potential significance of this work seems potentially high.

**Weaknesses:**

Overall, I expect readers won't have any major concerns with the theoretical results or experiments (see some minor questions below in the Questions section).

Rather, the main threat to the significance of this paper is making the case that that a Cournot-style is actually common in the UGC platforms being invoked here. Of course, even if only a few platforms really end up being well-described by the model, the contribution is still very meaningful. That said, a few specific concerns with the current draft:
- a number of specific platforms are mentioned by name: YouTube, TikTok, Netflix, Spotify, and MovieLens.
- Only data from MovieLens is used (which is very reasonable -- it's a very popular dataset for academic work for good reason).
- However, the named platforms vary quite a bit in terms of their actual creator competition, i.e. one would expect the incentives of a platform like Netflix (which also acts a creator agent, sometimes with substantially higher budget than other creators) to differ quite a bit from TikTok

See "Questions" section below for some specific questions about this concern that I think are likely to be in scope of a revision.

With this critique in mind -- that certain platforms might violate the assumptions needed for the model to work well -- I think the current draft may overstate the generality of the conceptual insight.

**Questions:**

A few very specific questions about the model (with the caveat that of course anything along the lines of using empirical data from major platforms and/or trying to frame this model as predictive for an entire spectrum of platform types is probably out of scope)
- What is the strongest evidence that any of these major platforms follow Cournot competition like dynamics?
- What is the impact of platform-as-creator dynamics, such on Netflix?
- More generally, it would be helpful to explicitly state how resource heterogeneity amongst creators or budget heterogeneity amongst consumers may or may not cause issues for the use of this model.
- To what extent would we expect results to hold if we did have access to MovieLens-style observational data from e.g. YouTube?

Overall, these are not "existential" questions per se, but some attempt to clarify could strengthen the draft quite a bit.

**Limitations:**

I do think the current draft could do more to justify the strength of the conceptual claims and/or hold a bit more space to explicitly discuss limitations (primarily, how well requisite assumptions hold across the platforms of interest). See above (Questions).

---

> ### Author Rebuttal · Authors · 2024-08-07
>
> We greatly appreciate the reviewer's suggestion for additional justification of our problem setting. Below, we respond to the questions raised.
>
> **Evidence of volume competition**
> Major UGC platforms, such as YouTube, TikTok, and Instagram, primarily generate profits through ad impressions, which are directly correlated with content impressions. Given this revenue model, it is logical that platform incentives are designed to encourage creators to compete for more impressions. By fostering an environment that motivates creators to increase their content visibility, platforms can maximize their ad revenue. This setup inherently triggers Cournot competition-like dynamics in UGC platforms, where creators compete for audience attention by producing more content.
>
> Our work is done in collaboration with a world-leading content recommendation and sharing platform with billions of users. We do observe Cournot competition-like dynamics from offline data analysis, and we will include such evidence in the revision once authorized by the company to disclose this information. Generally, we observed two phenomena:
> - Content embeddings from the same creators are clustered around a small region, indicating that most creators consistently produce content within the same topic.
> - Creator productivity is significantly influenced by the platform’s incentivizing plan; when the platform allocates more user traffic to a group of creators, the average production frequency of this group increases.
>
> These observations motivate the formulation of the $C^4$ framework.
>
> **How our model handles heterogeneity** We do allow for heterogeneity in budgets, as the cost function can be creator-dependent. For the discussion of heterogeneity in terms of content creation, please refer to our common response above.
>
> **How general is our model to different platforms** Our $C^4$ framework is applicable to mature content recommendation and sharing platforms where creators are aware of their expertise, consistently produce within their niche, and strategically adjust their production frequency to balance gain and cost. As discussed in lines 36-41, many existing leading platforms, such as YouTube and Instagram, exhibit these characteristics. On these platforms, most influential creators have clear and consistent branding and target specific user interest groups. Consequently, the platform’s recommendation strategy primarily impacts their willingness to engage, i.e., their content production frequency.
> However, our framework is less applicable to platforms like Netflix, where a different platform-as-creator dynamics prevail. The dynamics under such an environment is also interesting but is beyond the scope of our work. We have clarified this scope in the first paragraph of the introduction (see line 18), where we exclude non-UGC platforms like Netflix from our discussion.

---

> > ### Comment · Reviewer_1Tuv · 2024-08-08
> >
> > Thanks to the authors for this additional information.
> >
> > In my original review, I stated the view that even absent specific evidence about a particular UGC platform that the framework contribution of this work could be a meaningful reason to accept the paper.
> >
> > IMO the addition of offline data from a major platform will really boost the impact of the paper. While I understand it seems there is a still a chance the data cannot be shared, this reflects positively on the work.
> >
> > Beyond the potential of adding specific justifying data, the common response was also very helpful in clarifying concerns shared among reviewers.

---

> > > ### Author Response · Authors · 2024-08-11
> > > **Re: Official Comment by Reviewer 1Tuv**
> > >
> > > We sincerely thank Reviewer 1Tuv for their positive evaluation of our work and response. We particularly appreciate your accurate understanding of the contributions and potential of our research. We are currently working on providing an anonymized version of the offline data used to support our model, which will be included in the next version as additional justifying data. If you have any further questions, we would be more than happy to engage in further discussion.

---

### Official Review · Reviewer_1ynk · 2024-07-10

**Soundness:** 2
**Presentation:** 3
**Contribution:** 2
**Rating:** 4
**Confidence:** 3

**Summary:**

This paper studies the problem of the tradeoff between users’ satisfaction and creators’ engagement. Authors first define the traffic competition of creators on user-generated content platforms as a Cournot Content Creation Competition (C4) and establish corresponding PNEs. Based on PNEs, this work identifies the tradeoff between users’ and creators’ engagement and proposes the offline optimization solution to achieve the maximum social welfare by adjusting the exploration level of matching. Theoretical and empirical results are provided to support the effectiveness.

**Strengths:**

1.	Authors theoretically model the traffic competition among content creators as a C4 game, identify the tradeoff of user and creator engagement based on their theory, and finally find the optimal platform intervention to maximize the social welfare with the optimization method. Necessary proofs are provided with details.

2.	Based on the synthetic and real-world datasets, authors validate the phenomenon of the user-creator tradeoff (Figure 1) and the benefit of optimizing \beta (Figure 2). Authors also provide the results in Appendix with different $\lambda$ in the objective $W_\lambda$ to investigate the sensitivity of their solution when the target is changed.

3.	The manuscript is well-organized and easy to follow.

**Weaknesses:**

1.	Some assumptions are too strong, including (a) basic setups: Creators are producing contents with the same frequency and the same cost (only relate to the frequency) all the time. (b) platform intervention: all users contribute one unit amount of traffic, neglecting the dominant position of active users.
2.	Although the effectiveness is guaranteed by the theory and empirical study on small datasets, authors should also present the potential of the solution to be applied in the practical scenarios, e.g., how is the efficiency of the optimization, how to conduct the daily update of intervention strategy.
3.	Existing works have studied the C3 game. Authors may declare their unique contribution and improvement by considering “Cournot Competition” in their theory establishment and compare with previous methods in empirical validation.

**Questions:**

1.	How is the efficiency of the proposed solution when the dataset includes millions of users?
2.	What are the differences between C4 and previous C3 game? How does C4 benefit from the additional “Cournot” setting?

**Limitations:**

1.	Limited practical value. The assumption is too strong, and the experiments are constrained on small dataset with 1,000 users.
2.	Unclear distinct contributions compared with previous works.

---

> ### Author Rebuttal · Authors · 2024-08-07
>
> We thank the reviewer for pointing out the concerns and raising the clarification questions and we respond to them below accordingly.
>
> **Response to weakness 1: unit user traffic** First, we have to clarify that our model does not assume each user contributes only one unit of traffic. Since we do not impose any assumptions on the user distribution, we can easily represent active users by incorporating multiple instances of such a user in the formulation of creator utility function. For concerns regarding the assumption of homogeneous creation, please refer to our common response above.
>
> **Response to weakness 2 and question 1: computational efficiency** Thank you for these great suggestions. We discussed computational efficiency in detail in Section 6.2, providing a computational complexity upper bound of $O(n^2m + nm^2 + m^3)$, where one can cluster users and creators to share parameters within the same cluster ($n$ and $m$ represent the number of clusters for creators and users, respectively). By appropriately setting $n$ and $m$, we can tune the resolution of clustering and balance between the accuracy and efficiency of the optimization process described in Algorithm 1. Moreover, this optimization process does not need to be run frequently (e.g., on a daily basis), as every time Algorithm 1 identifies the optimal treatment for each user group based on the current user and creator preference distribution. It is only necessary to rerun the algorithm if there is a significant change in the global user or creator preference distribution. Therefore, the intervention strategy computed by Algorithm 1 does have strong potential for practical applications.
>
> **Response to weakness 3 and question 2: Difference from $C^3$ game** We highlighted the main differences between $C^4$ and $C^3$ in lines 32-41. First of all, we clarify that $C^4$ is NOT an extension of $C^3$, despite the similarity between their names. In fact, these two models correspond to entirely different competitive environments and assumptions about creator behavior. In $C^3$, creators have fixed creation frequencies and their strategy is to choose topics for producing content. In contrast, in $C^4$, creators' content has a stationary topic distribution, and their strategy is to determine the creation frequency. These two settings represent different stages of online content sharing platforms: $C^3$ models the early stage when creators are new to the platform and explore various possibilities to find their identity, while $C^4$ models a mature stage where creators are aware of their expertise and consistently produce within their niche, strategically adjusting their production frequency to balance gain and cost. Therefore, the formulation of $C^4$ addresses a very different practical problem and should be considered a complementary setting to $C^3$, and the results obtained in the previous $C^3$ setting do not provide any analytical insight for $C^4$.
>
> **Response to limitation 1: experiment scale** As explained in Section 6.2, the number of users is not the bottleneck for the scalability of our algorithm because we can cluster the user population and update $\beta$ at the group level (this is also a practical treatment in large-scale real-world systems). Therefore, we believe that our experiment with 1,000 users is representative and should be sufficient for larger-scale experiments.

---

> ### Author Response · Authors · 2024-08-11
> **Reminder to Reviewer 1ynk**
>
> Dear Reviewer 1ynk, as the discussion phase is coming to a close, we wanted to gently remind you to review our response and consider reevaluating our paper in light of the additional information provided. If you have any further suggestions or concerns, we would be more than happy to engage in further discussion to improve our work.

---

### Official Review · Reviewer_GVPh · 2024-07-16

**Soundness:** 3
**Presentation:** 3
**Contribution:** 2
**Rating:** 6
**Confidence:** 3

**Summary:**

The authors propose a new game-theoretical model Cournot Content Creation Competition ($C^4$), that studies the relation between the matching strategy of user-generated content (UGC) platforms and the production willingness of the platform’s content creators. Under certain assumptions, the authors show that the game has a unique Pure Nash equilibrium, and show that increasing matching accuracy elevates user satisfaction but also decreases the overall volume of content creation. Building on this tradeoff, the authors propose an optimization approach that balances the two objectives, providing both theoretical analysis and empirical simulations.

**Strengths:**

- Overall well-written.
- Interesting insight on the tradeoff between user satisfaction and creator engagement shown by theoretical analysis.

**Weaknesses:**

- (Main) Model might be too simplistic - authors assume users consistent produce work of same topic & quality and only changes the production volume.

- (Minor) The authors associate user satisfaction as a short-term goal for the platform and creation volume as a long-term goal for the platform. The authors make an argument for this in line 188-197, although I’m still not fully convinced:
   - The main imbalance that I feel comes from the fact that when I think of *long-term* goals of a platform, it's fundamentally intertwined with the ability for platforms to attract new and keep existing users, which comes from a user standpoint and not from a "content volume" standpoint. I get the authors argument when they mention how content creation frequency might harm user satisfaction (line 192 “users can hardly be satisfied by their previously consumed material”). However, given individual's limited attention span, I think this only happens when the number of creators are quite limited, and that it's unclear that a decrease of production frequency from say 2 weeks -> 3 weeks will result in a significant harm to the *long term* viability of a platform causing users to drop out in the long-term.
   - In general, this seems to point to an alternative model where the content volume *comes in* the user's utility model, where users’ utility are not only determined by how they liked the recommended content (which is the utility considered in the paper) but also by the availability of content on the platform, and they might drop out of the platform when their utility falls below a certain level. From this lens, it's less clear that this is a short-term v.s. long-term issue.

- Typo: i =1 -> j=1 in line 92, third and fourth -> second and fourth in line 347

**Questions:**

- Can the authors elaborate on the point on short v.s. long-term goals in the weakness section above? Specifically, why is user satisfaction short-term concern and content generation frequency long-term concern.
- In practical scenarios when an increase in content comes with quality degradations and topic differentials, are there intuitions on whether and under what scenarios the result can continues to hold?
- Is collusion between content creators a potential problem here?

**Limitations:**

The authors adequately addressed the limitations.

---

> ### Author Rebuttal · Authors · 2024-08-07
>
> We thank the reviewer for appreciating the insights of our problem setting and results. For our response to the suggested main weakness and question 2, please refer to our common response above. Below, we address the remaining questions.
>
> **Why total content volume affects long term user engagement** It is not a surprise that long-term user engagement on a platform is crucially dependent on the total content creation volume due to the following reasons.
>
> - First, Regular and abundant content updates provide users with a better experience of personalization, encouraging them to return to the platform more often or new users to join the platform, which increases engagement time and user satisfaction, and thus long-term success of the platform.
>
> - Even though active creators and users are usually large in number on leading UGC platforms, the creator-to-user ratio is typically small. For example, TikTok has billions of viewers but only around 1 million active creators. As a result, most users are highly loyal to specific creators and eagerly consume whatever they create. This user stickiness means that even a small increase in content creation frequency by these creators can significantly boost total user engagement. Therefore, while minor fluctuations in overall content creation frequency might not drastically affect user churn rates, a consistent decrease in content creation from active creators will have a significant negative impact on user engagement.
>
> In addition, we acknowledge that emphasizing the contrast between short-term and long-term goals might be a bit misleading. Our key point is to highlight the tradeoff between recommendation relevance and overall content creation volume in optimizing user engagement. Both factors are clearly important to the engagement target but we show that they are inherently conflicting when considering content creators' strategic responses. We will clarify this point in revision.
>
> **Alternative model that accounts for volume in utility** This is a very constructive and interesting idea, however, to justify such a user utility model that correlates with the volume of available content on the platform (even if they do not consume them), we need to introduce an ad-hoc user behavioral assumption describing how users' satisfaction or return to the platform  relates to the amount of available content. And oftentimes, ordinary users do not even know the amount of available content on a particular platform (think about YouTube and Instagram),  although a positive correlation might empirically exist. We recognize this as an intriguing direction and intend to investigate it in future work.
>
> **Collusion among creators** Indeed, many creators today are not working as individuals but as representatives of a team (e.g., belonging to the same studio or society). In our model, a player can be regarded as a team rather than an individual for such situations. However, if the reviewer is referring to collusion among different players to game the system, this is beyond the scope of our current framework and we might need a new model for studying such an intriguing phenomenon.

---

> > ### Comment · Reviewer_GVPh · 2024-08-08
> > **Response**
> >
> > I thank the authors for their response. Most of my concerns are addressed, and I'm happy to keep my score, leaning towards acceptance of the paper.

---

> > > ### Author Response · Authors · 2024-08-11
> > > **Re: Response**
> > >
> > > We are pleased that our response has addressed the reviewer's main concerns. We are currently working on providing an anonymized version of the offline data to strengthen our model assumptions, which will be included in the next version as additional supporting results. If you have any further suggestions for improving our work, please feel free to share them, and we would be more than happy to engage in further discussion.

---

### Author Rebuttal · Authors · 2024-08-07

**Common response**

We appreciate the reviewers' overall positive evaluations about our work, especially the acknowledgement of the significance of the problem setting, novelty of our analysis and insights from our theoretical and empirical results. We are happy to integrate the reviewers’ suggestions for improving the current version. In the following, we first respond to a common question regarding our modeling assumption raised by reviewers, and then answer each reviewer’s questions separately.

**Assumption about homogeneous creation** Reviewers raised a common concern that our model relies on the assumption that creators consistently produce content of the same topic and quality, only varying the production volume. However, our setting is more nuanced than the reviewers suggested. Rather than assuming creators are homogeneous in their creations, our framework incorporates variances  in quality and topics by the random variable  $\epsilon_{ij}$, which reflects the uncertainty of the matching score between user-$i$ and creator-$j$'s content. This means each creator in our model can produce multiple pieces of content with varying topics and qualities. The key assumption is that the average matching score of these contents to each user does not depend on  the creation volume. Essentially, the content from a creator can be drawn from a fixed distribution, and the variance of this distribution does not significantly impact the creator's utility function as long as it is not too large (e.g., not to be flat), given we consider the expected traffic each creator collects.

Moreover, our data analysis on a world-leading content creation and sharing platform indicates that most creators tend to stick to a specific topic (e.g., reflecting their expertise or identity), meaning that their produced content is indeed drawn from a static and well-concentrated distribution. This is not a surprise as such a strategy helps them build brand identity and uniqueness among their followers and also maximizes their chances of being favored by the platform's algorithm. However, due to company policy, we are unable to disclose specific details but can only provide a qualitative discussion in lines 36-41. We can add more data analysis results regarding this finding to further support our modeling assumptions once we obtain the legal permission of the company.

---

### Decision · Program_Chairs · 2024-09-25

**Decision:**

Accept (poster)

**Comment:**

The paper investigates the dynamics between user-generated content (UGC) platforms and content creators, particularly focusing on the impact of recommendation algorithms on creator behavior. The authors propose a game-theoretical model, the Cournot Content Creation Competition (C4), to study the trade-offs between short-term user satisfaction and long-term content diversity. The paper introduces an optimization approach to balance these competing objectives, providing both theoretical insights and empirical validation using synthetic data and real-world datasets.

Generally, reviewers agree that the research question of modeling the content generation/consumption of UGC is well motivated and that the results are technically solid overall. Many of the questions and concerns raised revolve around the justifications of the models and objectives. We encourage authors take the reviewer comments into account in preparing their final version.

One additional comment that's raised during the discussion phase: The research question of modeling and analyzing UGC from game theory and learning perspectives has been explored for quite some time, however, most of the citations in this work are from the past few years. It would help if the authors can include the discussion to these earlier works to better contextualize their contributions. As examples, below are two of the papers along this line:

- Incentivizing High-quality User-Generated Content. Arpita Ghosh, Preston McAfee. WWW 2011.
- Learning and Incentives in User-Generated Content: Multi-Armed Bandits with Endogenous Arms. Arpita Ghosh, Patrick Hummel. ITCS 2013.